# Unsupervised Image-to-Image Translation Networks

**Ming-Yu Liu,   Thomas Breuel,   Jan Kautz**
NVIDIA
{mingyul,tbreuel,jkautz}@nvidia.com

## Abstract

Unsupervised image-to-image translation aims at learning a joint distribution of images in different domains by using images from the marginal distributions in individual domains. Since there exists an infinite set of joint distributions that can arrive the given marginal distributions, one could infer nothing about the joint distribution from the marginal distributions without additional assumptions. To address the problem, we make a shared-latent space assumption and propose an unsupervised image-to-image translation framework based on Coupled GANs. We compare the proposed framework with competing approaches and present high quality image translation results on various challenging unsupervised image translation tasks, including street scene image translation, animal image translation, and face image translation. We also apply the proposed framework to domain adaptation and achieve state-of-the-art performance on benchmark datasets. Code and additional results are available in https://github.com/mingyuliutw/unit.

## 1   Introduction

Many computer visions problems can be posed as an image-to-image translation problem, mapping an image in one domain to a corresponding image in another domain. For example, super-resolution can be considered as a problem of mapping a low-resolution image to a corresponding high-resolution image; colorization can be considered as a problem of mapping a gray-scale image to a corresponding color image. The problem can be studied in supervised and unsupervised learning settings. In the supervised setting, paired of corresponding images in different domains are available [8, 15]. In the unsupervised setting, we only have two independent sets of images where one consists of images in one domain and the other consists of images in another domain—there exist no paired examples showing how an image could be translated to a corresponding image in another domain. Due to lack of corresponding images, the UNsupervised Image-to-image Translation (UNIT) problem is considered harder, but it is more applicable since training data collection is easier.

When analyzing the image translation problem from a probabilistic modeling perspective, the key challenge is to learn a joint distribution of images in different domains. In the unsupervised setting, the two sets consist of images from two marginal distributions in two different domains, and the task is to infer the joint distribution using these images. The coupling theory [16] states there exist an infinite set of joint distributions that can arrive the given marginal distributions in general. Hence, inferring the joint distribution from the marginal distributions is a highly ill-posed problem. To address the ill-posed problem, we need additional assumptions on the structure of the joint distribution.

To this end we make a shared-latent space assumption, which assumes a pair of corresponding images in different domains can be mapped to a same latent representation in a shared-latent space. Based on the assumption, we propose a UNIT framework that are based on generative adversarial networks (GANs) and variational autoencoders (VAEs). We model each image domain using a VAE-GAN. The adversarial training objective interacts with a weight-sharing constraint, which enforces a shared-latent space, to generate corresponding images in two domains, while the variational autoencoders relate translated images with input images in the respective domains. We applied the proposed

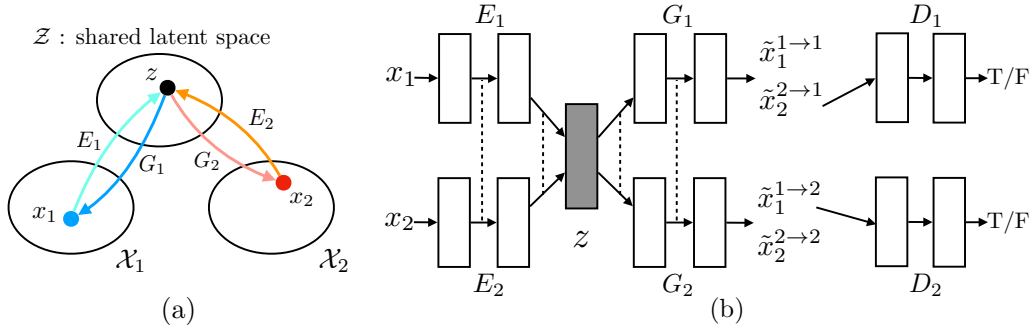

Figure 1: (a) The shared latent space assumption. We assume a pair of corresponding images $(x_1, x_2)$ in two different domains $\mathcal{X}_1$ and $\mathcal{X}_2$ can be mapped to a same latent code $z$ in a shared-latent space $\mathcal{Z}$. $E_1$ and $E_2$ are two encoding functions, mapping images to latent codes. $G_1$ and $G_2$ are two generation functions, mapping latent codes to images. (b) The proposed UNIT framework. We represent $E_1$ $E_2$ $G_1$ and $G_2$ using CNNs and implement the shared-latent space assumption using a weight sharing constraint where the connection weights of the last few layers (high-level layers) in $E_1$ and $E_2$ are tied (illustrated using dashed lines) and the connection weights of the first few layers (high-level layers) in $G_1$ and $G_2$ are tied. Here, $\tilde{x}_1^{1 \to 1}$ and $\tilde{x}_2^{2 \to 2}$ are self-reconstructed images, and $\tilde{x}_1^{1 \to 2}$ and $\tilde{x}_2^{2 \to 1}$ are domain-translated images. $D_1$ and $D_2$ are adversarial discriminators for the respective domains, in charge of evaluating whether the translated images are realistic.

Table 1: Interpretation of the roles of the subnetworks in the proposed framework.

| Networks | $\{E_1, G_1\}$ | $\{E_1, G_2\}$ | $\{G_1, D_1\}$ | $\{E_1, G_1, D_1\}$ | $\{G_1, G_2, D_1, D_2\}$ |
|---|---|---|---|---|---|
| Roles | VAE for $\mathcal{X}_1$ | Image Translator $\mathcal{X}_1 \to \mathcal{X}_2$ | GAN for $\mathcal{X}_1$ | VAE-GAN [14] | CoGAN [17] |

framework to various unsupervised image-to-image translation problems and achieved high quality image translation results. We also applied it to the domain adaptation problem and achieved state-of-the-art accuracies on benchmark datasets. The shared-latent space assumption was used in Coupled GAN [17] for joint distribution learning. Here, we extend the Coupled GAN work for the UNIT problem. We also note that several contemporary works propose the cycle-consistency constraint assumption [29, 10], which hypothesizes the existence of a cycle-consistency mapping so that an image in the source domain can be mapped to an image in the target domain and this translated image in the target domain can be mapped back to the original image in the source domain. In the paper, we show that the shared-latent space constraint implies the cycle-consistency constraint.

## 2 Assumptions

Let $\mathcal{X}_1$ and $\mathcal{X}_2$ be two image domains. In supervised image-to-image translation, we are given samples $(x_1, x_2)$ drawn from a joint distribution $P_{\mathcal{X}_1, \mathcal{X}_2}(x_1, x_2)$. In unsupervised image-to-image translation, we are given samples drawn from the marginal distributions $P_{\mathcal{X}_1}(x_1)$ and $P_{\mathcal{X}_2}(x_2)$. Since an infinite set of possible joint distributions can yield the given marginal distributions, we could infer nothing about the joint distribution from the marginal samples without additional assumptions.

We make the shared-latent space assumption. As shown Figure 1, we assume for any given pair of images $x_1$ and $x_2$, there exists a shared latent code $z$ in a shared-latent space, such that we can recover both images from this code, and we can compute this code from each of the two images. That is, we postulate there exist functions $E_1^*$, $E_2^*$, $G_1^*$, and $G_2^*$ such that, given a pair of corresponding images $(x_1, x_2)$ from the joint distribution, we have $z = E_1^*(x_1) = E_2^*(x_2)$ and conversely $x_1 = G_1^*(z)$ and $x_2 = G_2^*(z)$. Within this model, the function $x_2 = F_{1 \to 2}^*(x_1)$ that maps from $\mathcal{X}_1$ to $\mathcal{X}_2$ can be represented by the composition $F_{1 \to 2}^*(x_1) = G_2^*(E_1^*(x_1))$. Similarly, $x_1 = F_{2 \to 1}^*(x_2) = G_1^*(E_2^*(x_2))$. The UNIT problem then becomes a problem of learning $F_{1 \to 2}^*$ and $F_{2 \to 1}^*$. We note that a necessary condition for $F_{1 \to 2}^*$ and $F_{2 \to 1}^*$ to exist is the cycle-consistency constraint [29, 10]: $x_1 = F_{2 \to 1}^*(F_{1 \to 2}^*(x_1))$ and $x_2 = F_{1 \to 2}^*(F_{2 \to 1}^*(x_2))$. We can reconstruct the input image from translating back the translated input image. In other words, the proposed shared-latent space assumption implies the cycle-consistency assumption (but not vice versa).

To implement the shared-latent space assumption, we further assume a shared intermediate representation $h$ such that the process of generating a pair of corresponding images admits a form of

$$z \to h \Big\langle \begin{array}{c} x_1 \\ x_2 \end{array} . \tag{1}$$

Consequently, we have $G_1^* \equiv G_{L,1}^* \circ G_H^*$ and $G_2^* \equiv G_{L,2}^* \circ G_H^*$ where $G_H^*$ is a common high-level generation function that maps $z$ to $h$ and $G_{L,1}^*$ and $G_{L,2}^*$ are low-level generation functions that map $h$ to $x_1$ and $x_2$, respectively. In the case of multi-domain image translation (e.g., sunny and rainy image translation), $z$ can be regarded as the compact, high-level representation of a scene ("car in front, trees in back"), and $h$ can be considered a particular realization of $z$ through $G_H^*$ ("car/tree occupy the following pixels"), and $G_{L,1}^*$ and $G_{L,2}^*$ would be the actual image formation functions in each modality ("tree is lush green in the sunny domain, but dark green in the rainy domain"). Assuming $h$ also allow us to represent $E_1^*$ and $E_2^*$ by $E_1^* \equiv E_H^* \circ E_{L,1}^*$ and $E_2^* \equiv E_H^* \circ E_{L,2}^*$.

In the next section, we discuss how we realize the above ideas in the proposed UNIT framework.

## 3   Framework

Our framework, as illustrated in Figure 1, is based on variational autoencoders (VAEs) [13, 22, 14] and generative adversarial networks (GANs) [6, 17]. It consists of 6 subnetworks: including two domain image encoders $E_1$ and $E_2$, two domain image generators $G_1$ and $G_2$, and two domain adversarial discriminators $D_1$ and $D_2$. Several ways exist to interpret the roles of the subnetworks, which we summarize in Table 1. Our framework learns translation in both directions in one shot.

**VAE.** The encoder–generator pair $\{E_1, G_1\}$ constitutes a VAE for the $\mathcal{X}_1$ domain, termed VAE$_1$. For an input image $x_1 \in \mathcal{X}_1$, the VAE$_1$ first maps $x_1$ to a code in a latent space $\mathcal{Z}$ via the encoder $E_1$ and then decodes a random-perturbed version of the code to reconstruct the input image via the generator $G_1$. We assume the components in the latent space $\mathcal{Z}$ are conditionally independent and Gaussian with unit variance. In our formulation, the encoder outputs a mean vector $E_{\mu,1}(x_1)$ and the distribution of the latent code $z_1$ is given by $q_1(z_1|x_1) \equiv \mathcal{N}(z_1|E_{\mu,1}(x_1), I)$ where $I$ is an identity matrix. The reconstructed image is $\tilde{x}_1^{1\to1} = G_1(z_1 \sim q_1(z_1|x_1))$. Note that here we abused the notation since we treated the distribution of $q_1(z_1|x_1)$ as a random vector of $\mathcal{N}(E_{\mu,1}(x_1), I)$ and sampled from it. Similarly, $\{E_2, G_2\}$ constitutes a VAE for $\mathcal{X}_2$: VAE$_2$ where the encoder $E_2$ outputs a mean vector $E_{\mu,2}(x_2)$ and the distribution of the latent code $z_2$ is given by $q_2(z_2|x_2) \equiv \mathcal{N}(z_2|E_{\mu,2}(x_2), I)$. The reconstructed image is $\tilde{x}_2^{2\to2} = G_2(z_2 \sim q_2(z_2|x_2))$.

Utilizing the reparameterization trick [13], the non-differentiable sampling operation can be reparameterized as a differentiable operation using auxiliary random variables. This reparameterization trick allows us to train VAEs using back-prop. Let $\eta$ be a random vector with a multi-variate Gaussian distribution: $\eta \sim \mathcal{N}(\eta|0, I)$. The sampling operations of $z_1 \sim q_1(z_1|x_1)$ and $z_2 \sim q_2(z_2|x_2)$ can be implemented via $z_1 = E_{\mu,1}(x_1) + \eta$ and $z_2 = E_{\mu,2}(x_2) + \eta$, respectively.

**Weight-sharing.** Based on the shared-latent space assumption discussed in Section 2, we enforce a weight-sharing constraint to relate the two VAEs. Specifically, we share the weights of the last few layers of $E_1$ and $E_2$ that are responsible for extracting high-level representations of the input images in the two domains. Similarly, we share the weights of the first few layers of $G_1$ and $G_2$ responsible for decoding high-level representations for reconstructing the input images.

Note that the weight-sharing constraint alone does not guarantee that corresponding images in two domains will have the same latent code. In the unsupervised setting, no pair of corresponding images in the two domains exists to train the network to output a same latent code. The extracted latent codes for a pair of corresponding images are different in general. Even if they are the same, the same latent component may have different semantic meanings in different domains. Hence, the same latent code could still be decoded to output two unrelated images. However, we will show that through adversarial training, a pair of corresponding images in the two domains can be mapped to a common latent code by $E_1$ and $E_2$, respectively, and a latent code will be mapped to a pair of corresponding images in the two domains by $G_1$ and $G_2$, respectively.

The shared-latent space assumption allows us to perform image-to-image translation. We can translate an image $x_1$ in $\mathcal{X}_1$ to an image in $\mathcal{X}_2$ through applying $G_2(z_1 \sim q_1(z_1|x_1))$. We term such an information processing stream as the image translation stream. Two image translation streams exist in the proposed framework: $\mathcal{X}_1 \to \mathcal{X}_2$ and $\mathcal{X}_2 \to \mathcal{X}_1$. The two streams are trained jointly with the two image reconstruction streams from the VAEs. Once we could ensure that a pair of corresponding

images are mapped to a same latent code and a same latent code is decoded to a pair of corresponding images, $(x_1, G_2(z_1 \sim q_1(z_1|x_1)))$ would form a pair of corresponding images. In other words, the composition of $E_1$ and $G_2$ functions approximates $F_{1 \to 2}^*$ for unsupervised image-to-image translation discussed in Section 2, and the composition of $E_2$ and $G_1$ function approximates $F_{2 \to 1}^*$.

**GANs.** Our framework has two generative adversarial networks: $\text{GAN}_1 = \{D_1, G_1\}$ and $\text{GAN}_2 = \{D_2, G_2\}$. In $\text{GAN}_1$, for real images sampled from the first domain, $D_1$ should output true, while for images generated by $G_1$, it should output false. $G_1$ can generate two types of images: 1) images from the reconstruction stream $\tilde{x}_1^{1 \to 1} = G_1(z_1 \sim q_1(z_1|x_1))$ and 2) images from the translation stream $\tilde{x}_2^{2 \to 1} = G_1(z_2 \sim q_2(z_2|x_2))$. Since the reconstruction stream can be supervisedly trained, it is suffice that we only apply adversarial training to images from the translation stream, $\tilde{x}_2^{2 \to 1}$. We apply a similar processing to $\text{GAN}_2$ where $D_2$ is trained to output true for real images sampled from the second domain dataset and false for images generated from $G_2$.

**Cycle-consistency (CC).** Since the shared-latent space assumption implies the cycle-consistency constraint (See Section 2), we could also enforce the cycle-consistency constraint in the proposed framework to further regularize the ill-posed unsupervised image-to-image translation problem. The resulting information processing stream is called the cycle-reconstruction stream.

**Learning.** We jointly solve the learning problems of the $\text{VAE}_1$, $\text{VAE}_2$, $\text{GAN}_1$ and $\text{GAN}_2$ for the image reconstruction streams, the image translation streams, and the cycle-reconstruction streams:

$$\min_{E_1, E_2, G_1, G_2} \max_{D_1, D_2} \mathcal{L}_{\text{VAE}_1}(E_1, G_1) + \mathcal{L}_{\text{GAN}_1}(E_1, G_1, D_1) + \mathcal{L}_{\text{CC}_1}(E_1, G_1, E_2, G_2)$$
$$\mathcal{L}_{\text{VAE}_2}(E_2, G_2) + \mathcal{L}_{\text{GAN}_2}(E_2, G_2, D_2) + \mathcal{L}_{\text{CC}_2}(E_2, G_2, E_1, G_1). \quad (2)$$

VAE training aims for minimizing a variational upper bound In (2), the VAE objects are

$$\mathcal{L}_{\text{VAE}_1}(E_1, G_1) = \lambda_1 \text{KL}(q_1(z_1|x_1)||p_\eta(z)) - \lambda_2 \mathbb{E}_{z_1 \sim q_1(z_1|x_1)}[\log p_{G_1}(x_1|z_1)] \quad (3)$$
$$\mathcal{L}_{\text{VAE}_2}(E_2, G_2) = \lambda_1 \text{KL}(q_2(z_2|x_2)||p_\eta(z)) - \lambda_2 \mathbb{E}_{z_2 \sim q_2(z_2|x_2)}[\log p_{G_2}(x_2|z_2)]. \quad (4)$$

where the hyper-parameters $\lambda_1$ and $\lambda_2$ control the weights of the objective terms and the KL divergence terms penalize deviation of the distribution of the latent code from the prior distribution. The regularization allows an easy way to sample from the latent space [13]. We model $p_{G_1}$ and $p_{G_2}$ using Laplacian distributions, respectively. Hence, minimizing the negative log-likelihood term is equivalent to minimizing the absolute distance between the image and the reconstructed image. The prior distribution is a zero mean Gaussian $p_\eta(z) = \mathcal{N}(z|0, I)$.

In (2), the GAN objective functions are given by

$$\mathcal{L}_{\text{GAN}_1}(E_1, G_1, D_1) = \lambda_0 \mathbb{E}_{x_1 \sim P_{\mathcal{X}_1}}[\log D_1(x_1)] + \lambda_0 \mathbb{E}_{z_2 \sim q_2(z_2|x_2)}[\log(1 - D_1(G_1(z_2)))] \quad (5)$$
$$\mathcal{L}_{\text{GAN}_2}(E_2, G_2, D_2) = \lambda_0 \mathbb{E}_{x_2 \sim P_{\mathcal{X}_2}}[\log D_2(x_2)] + \lambda_0 \mathbb{E}_{z_1 \sim q_1(z_1|x_1)}[\log(1 - D_2(G_2(z_1)))]. \quad (6)$$

The objective functions in (5) and (6) are conditional GAN objective functions. They are used to ensure the translated images resembling images in the target domains, respectively. The hyper-parameter $\lambda_0$ controls the impact of the GAN objective functions.

We use a VAE-like objective function to model the cycle-consistency constraint, which is given by

$$\mathcal{L}_{\text{CC}_1}(E_1, G_1, E_2, G_2) = \lambda_3 \text{KL}(q_1(z_1|x_1)||p_\eta(z)) + \lambda_3 \text{KL}(q_2(z_2|x_1^{1 \to 2}))||p_\eta(z)) -$$
$$\lambda_4 \mathbb{E}_{z_2 \sim q_2(z_2|x_1^{1 \to 2})}[\log p_{G_1}(x_1|z_2)] \quad (7)$$
$$\mathcal{L}_{\text{CC}_2}(E_2, G_2, E_1, G_1) = \lambda_3 \text{KL}(q_2(z_2|x_2)||p_\eta(z)) + \lambda_3 \text{KL}(q_1(z_1|x_2^{2 \to 1}))||p_\eta(z)) -$$
$$\lambda_4 \mathbb{E}_{z_1 \sim q_1(z_1|x_2^{2 \to 1})}[\log p_{G_2}(x_2|z_1)]. \quad (8)$$

where the negative log-likelihood objective term ensures a twice translated image resembles the input one and the KL terms penalize the latent codes deviating from the prior distribution in the cycle-reconstruction stream (Therefore, there are two KL terms). The hyper-parameters $\lambda_3$ and $\lambda_4$ control the weights of the two different objective terms.

Inheriting from GAN, training of the proposed framework results in solving a mini-max problem where the optimization aims to find a saddle point. It can be seen as a two player zero-sum game. The first player is a team consisting of the encoders and generators. The second player is a team consisting of the adversarial discriminators. In addition to defeating the second player, the first player has to minimize the VAE losses and the cycle-consistency losses. We apply an alternating gradient

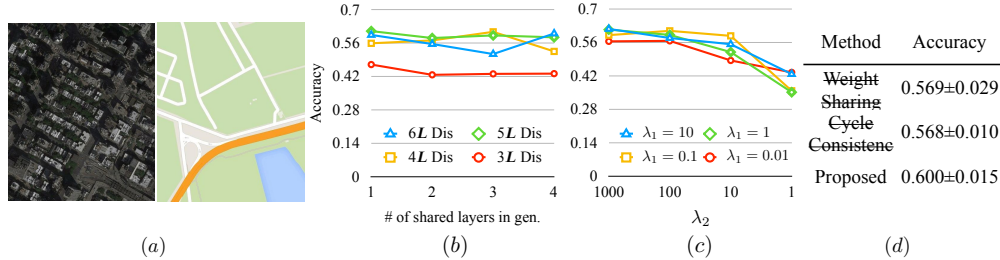

|  | Method | Accuracy |
|---|---|---|
|  | Weight Sharing | 0.569±0.029 |
|  | Cycle Consistenc | 0.568±0.010 |
|  | Proposed | 0.600±0.015 |

$(a)$          $(b)$          $(c)$          $(d)$

Figure 2: (a) Illustration of the Map dataset. Left: satellite image. Right: map. We translate holdout satellite images to maps and measure the accuracy achieved by various configurations of the proposed framework. (b) Translation accuracy versus different network architectures. (c) Translation accuracy versus different hyper-parameter values. (d) Impact of weight-sharing and cycle-consistency constraints on translation accuracy.

update scheme similar to the one described in [6] to solve (2). Specifically, we first apply a gradient ascent step to update $D_1$ and $D_2$ with $E_1$, $E_2$, $G_1$, and $G_2$ fixed. We then apply a gradient descent step to update $E_1$, $E_2$, $G_1$, and $G_2$ with $D_1$ and $D_2$ fixed.

**Translation:** After learning, we obtain two image translation functions by assembling a subset of the subnetworks. We have $F_{1\to2}(x_1) = G_2(z_1 \sim q_1(z_1|x_1))$ for translating images from $\mathcal{X}_1$ to $\mathcal{X}_2$ and $F_{2\to1}(x_2) = G_1(z_2 \sim q_2(z_2|x_2))$ for translating images from $\mathcal{X}_2$ to $\mathcal{X}_1$.

## 4 Experiments

We first analyze various components of the proposed framework. We then present visual results on challenging translation tasks. Finally, we apply our framework to the domain adaptation tasks.

**Performance Analysis.** We used ADAM [11] for training where the learning rate was set to 0.0001 and momentums were set to 0.5 and 0.999. Each mini-batch consisted of one image from the first domain and one image from the second domain. Our framework had several hyper-parameters. The default values were $\lambda_0 = 10$, $\lambda_3 = \lambda_1 = 0.1$ and $\lambda_4 = \lambda_2 = 100$. For the network architecture, our encoders consisted of 3 convolutional layers as the front-end and 4 basic residual blocks [7] as the back-end. The generators consisted of 4 basic residual blocks as the front-end and 3 transposed convolutional layers as the back-end. The discriminators consisted of stacks of convolutional layers. We used LeakyReLU for nonlinearity. The details of the networks are given in Appendix A.

We used the map dataset [8] (visualized in Figure 2), which contained corresponding pairs of images in two domains (satellite image and map) useful for quantitative evaluation. Here, the goal was to learn to translate between satellite images and maps. We operated in an unsupervised setting where we used the 1096 satellite images from the training set as the first domain and 1098 maps from the validation set as the second domain. We trained for 100K iterations and used the final model to translate 1098 satellite images in the test set. We then compared the difference between a translated satellite image (supposed to be maps) and the corresponding ground truth maps pixel-wisely. A pixel translation was counted correct if the color difference was within 16 of the ground truth color value. We used the average pixel accuracy over the images in the test set as the performance metric. We could use color difference for measuring translation accuracy since the target translation function was unimodal. We did not evaluate the translation from maps to images since the translation was multi-modal, which was difficult to construct a proper evaluation metric.

In one experiment, we varied the number of weight-sharing layers in the VAEs and paired each configuration with discriminator architectures of different depths during training. We changed the number of weight-sharing layers from 1 to 4. (Sharing 1 layer in VAEs means sharing 1 layer for $E_1$ and $E_2$ and, at the same time, sharing 1 layer for $G_1$ and $G_2$.) The results were reported in Figure 2(b). Each curve corresponded to a discriminator architecture of a different depth. The x-axis denoted the number of weigh-sharing layers in the VAEs. We found that the shallowest discriminator architecture led to the worst performance. We also found that the number of weight-sharing layer had little impact. This was due to the use of the residual blocks. As tying the weight of one layer, it effectively constrained the other layers since the residual blocks only updated the residual information. In the rest of the experiments, we used VAEs with 1 sharing layer and discriminators of 5 layers.

We analyzed impact of the hyper-parameter values to the translation accuracy. For different weight values on the negative log likelihood terms (i.e., $\lambda_2$, $\lambda_4$), we computed the achieved translation accuracy over different weight values on the KL terms (i.e., $\lambda_1$, $\lambda_3$). The results were reported in Figure 2(c). We found that, in general, a larger weight value on the negative log likelihood terms yielded a better translation accuracy. We also found setting the weights of the KL terms to 0.1 resulted in consistently good performance. We hence set $\lambda_1 = \lambda_3 = 0.1$ and $\lambda_2 = \lambda_4 = 100$.

We performed an ablation study measuring impact of the weight-sharing and cycle-consistency constraints to the translation performance and showed the results in Figure 2(d). We reported average accuracy over 5 trials (trained with different initialized weights.). We note that when we removed the weight-sharing constraint (as a consequence, we also removed the reconstruction streams in the framework), the framework was reduced to the CycleGAN architecture [29, 10]. We found the model achieved an average pixel accuracy of 0.569. When we removed the cycle-consistency constraint and only used the weight-sharing constraint[1], it achieved 0.568 average pixel accuracy. But when we used the full model, it achieved the best performance of 0.600 average pixel accuracy. This echoed our point that for the ill-posed joint distribution recovery problem, more constraints are beneficial.

**Qualitative results.** Figure 3 to 6 showed results of the proposed framework on various UNIT tasks.

*Street images.* We applied the proposed framework to several unsupervised street scene image translation tasks including sunny to rainy, day to night, summery to snowy, and vice versa. For each task, we used a set of images extracted from driving videos recorded at different days and cities. The numbers of the images in the sunny/day, rainy, night, summery, and snowy sets are 86165, 28915, 36280, 6838, and 6044. We trained the network to translate street scene image of size 640×480. In Figure 3, we showed several example translation results . We found that our method could generate realistic translated images. We also found that one translation was usually harder than the other. Specifically, the translation that required adding more details to the image was usually harder (e.g. night to day). Additional results are available in https://github.com/mingyuliutw/unit.

*Synthetic to real.* In Figure 3, we showed several example results achieved by applying the proposed framework to translate images between the synthetic images in the SYNTHIA dataset [23] and the real images in the Cityscape dataset [2]. For the real to synthetic translation, we found our method made the cityscape images cartoon like. For the synthetic to real translation, our method achieved better results in the building, sky, road, and car regions than in the human regions.

*Dog breed conversion.* We used the images of Husky, German Shepherd, Corgi, Samoyed, and Old English Sheep dogs in the ImageNet dataset to learn to translate dog images between different breeds. We only used the head regions, which were extracted by a template matching algorithm. Several example results were shown in Figure 4. We found our method translated a dog to a different breed.

*Cat species conversion.* We also used the images of house cat, tiger, lion, cougar, leopard, jaguar, and cheetah in the ImageNet dataset to learn to translate cat images between different species. We only used the head regions, which again were extracted by a template matching algorithm. Several example results were shown in Figure 5. We found our method translated a cat to a different specie.

*Face attribute.* We used the CelebA dataset [18] for attribute-based face images translation. Each face image in the dataset had several attributes, including blond hair, smiling, goatee, and eyeglasses. The face images with an attribute constituted the 1st domain, while those without the attribute constituted the 2nd domain. In Figure 6, we visualized the results where we translated several images that do not have blond hair, eye glasses, goatee, and smiling to corresponding images with each of the individual attributes. We found that the translated face images were realistic.

**Domain Adaptation.** We applied the proposed framework to the problem for adapting a classifier trained using labeled samples in one domain (source domain) to classify samples in a new domain (target domain) where labeled samples in the new domain are unavailable during training. Early works have explored ideas from subspace learning [4] to deep feature learning [5, 17, 26].

We performed multi-task learning where we trained the framework to 1) translate images between the source and target domains and 2) classify samples in the source domain using the features extracted by the discriminator in the source domain. Here, we tied the weights of the high-level layers of $D_1$ and $D_2$. This allows us to adapt a classifier trained in the source domain to the target domain. Also, for a pair of generated images in different domains, we minimized the L1 distance

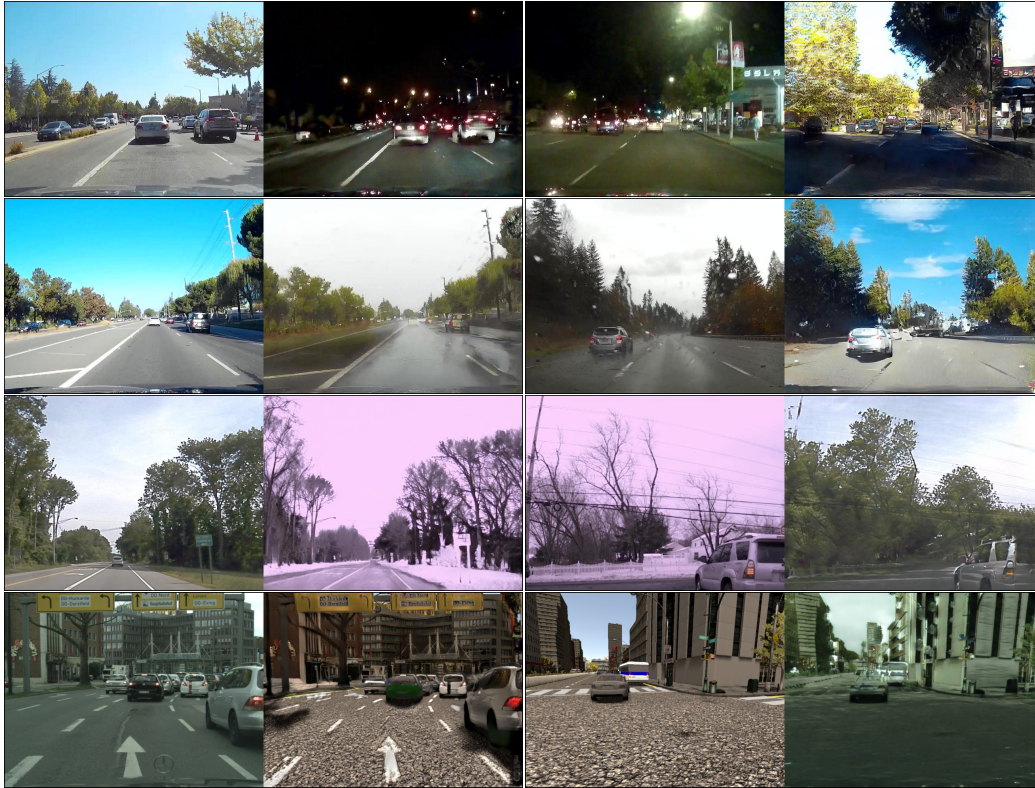

Figure 3: Street scene image translation results. For each pair, left is input and right is the translated image.

| Input | Old Eng. Sheep Dog | Husky | German Shepherd | Corgi | Input | Husky | Corgi |

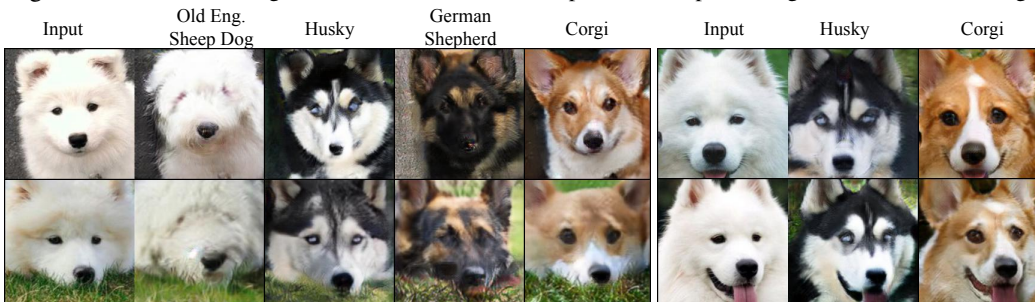

Figure 4: Dog breed translation results.

| Input | Cougar | Cheetah | Leopard | Lion | Tiger | Input | Leopard |

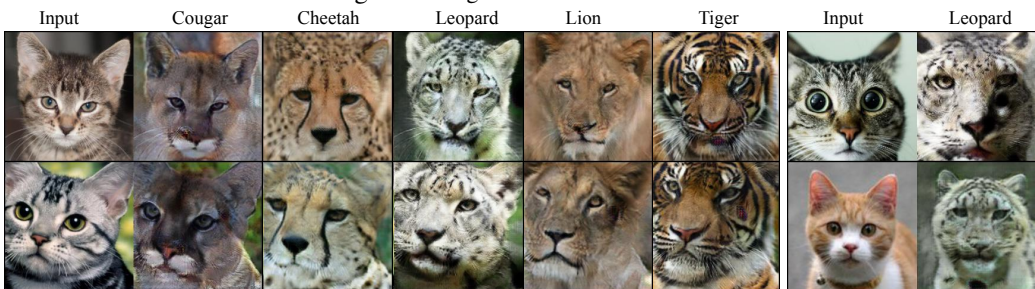

Figure 5: Cat species translation results.

| Input | +Blond Hair | +Eyeglasses | +Goatee | +Smiling | Input | +Blond Hair | +Eyeglasses | +Goatee | +Smiling |

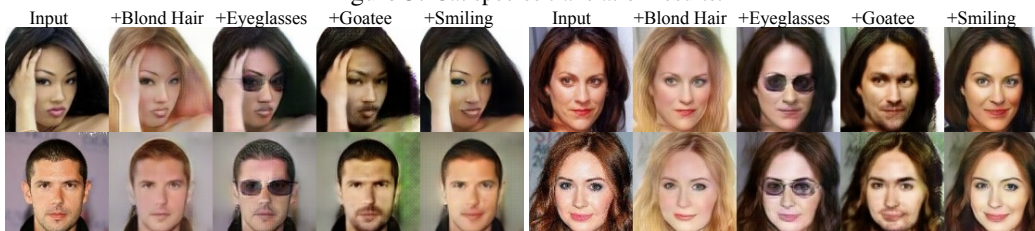

Figure 6: Attribute-based face translation results.

Table 2: Unsupervised domain adaptation performance. The reported numbers are classification accuracies.

| Method | SA [4] | DANN [5] | DTN [26] | CoGAN | UNIT (proposed) |
|---|---|---|---|---|---|
| SVHN→ MNIST | 0.5932 | 0.7385 | 0.8488 | - | **0.9053** |
| MNIST→ USPS | - | - | - | 0.9565 | **0.9597** |
| USPS→ MNIST | - | - | - | 0.9315 | **0.9358** |

between the features extracted by the highest layer of the discriminators, which further encouraged $D_1$ and $D_2$ to interpret a pair of corresponding images in the same way. We applied the approach to several tasks including adapting from the Street View House Number (SVHN) dataset [20] to the MNIST dataset and adapting between the MNIST and USPS datasets. Table 2 reported the achieved performance with comparison to the competing approaches. We found that our method achieved a 0.9053 accuracy for the SVHN→MNIST task, which was much better than 0.8488 achieved by the previous state-of-the-art method [26]. We also achieved better performance for the MNIST↔SVHN task than the Coupled GAN approach, which was the state-of-the-art. The digit images had a small resolution. Hence, we used a small network. We also found that the cycle-consistency constraint was not necessary for this task. More details about the experiments are available in Appendix B.

## 5 Related Work

Several deep generative models were recently proposed for image generation including GANs [6], VAEs [13, 22], and PixelCNN [27]. The proposed framework was based on GANs and VAEs but it was designed for the unsupervised image-to-image translation task, which could be considered as a conditional image generation model. In the following, we first review several recent GAN and VAE works and then discuss related image translation works.

**GAN** learning is via staging a zero-sum game between the generator and discriminator. The quality of GAN-generated images had improved dramatically since the introduction. LapGAN [3] proposed a Laplacian pyramid implementation of GANs. DCGAN [21] used a deeper convolutional network. Several GAN training tricks were proposed in [24]. WGAN [1] used the Wasserstein distance.

**VAEs** optimize a variational bound. By improving the variational approximation, better image generation results were achieved [19, 12]. In [14], a VAE-GAN architecture was proposed to improve image generation quality of VAEs. VAEs were applied to translate face image attribute in [28].

Conditional generative model is a popular approach for mapping an image from one domain to another. Most of the existing works were based on supervised learning [15, 8, 9]. Our work differed to the previous works in that we do not need corresponding images. Recently, [26] proposed the domain transformation network (DTN) and achieved promising results on translating small resolution face and digit images. In addition to faces and digits, we demonstrated that the proposed framework can translate large resolution natural images. It also achieved a better performance in the unsupervised domain adaptation task. In [25], a conditional generative adversarial network-based approach was proposed to translate a rendering images to a real image for gaze estimation. In order to ensure the generated real image was similar to the original rendering image, the L1 distance between the generated and original image was minimized. We note that two contemporary papers [29, 10] independently introduced the cycle-consistency constraint for the unsupervised image translation. We showed that that the cycle-consistency constraint is a natural consequence of the proposed shared-latent space assumption. From our experiment, we found that cycle-consistency and the weight-sharing (a realization of the shared-latent space assumption) constraints rendered comparable performance. When the two constraints were jointed used, the best performance was achieved.

## 6 Conclusion and Future Work

We presented a general framework for unsupervised image-to-image translation. We showed it learned to translate an image from one domain to another without any corresponding images in two domains in the training dataset. The current framework has two limitations. First, the translation model is unimodal due to the Gaussian latent space assumption. Second, training could be unstable due to the saddle point searching problem. We plan to address these issues in the future work.

## Footnotes

[1]We used this architecture in an earlier version of the paper.

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
