[Supplementary Material]

# A Network Architecture

The network architecture used for the unsupervised image-to-image translation experiments is given in Table 3. We use the following abbreviation for ease of presentation: N=Neurons, K=Kernel size, S=Stride size. The transposed convolutional layer is denoted by DCONV. The residual basic block is denoted as RESBLK.

Table 3: Network architecture for the unsupervised image translation experiments.

| Layer | Encoders | Shared? |
|---|---|---|
| 1 | CONV-(N64,K7,S2), LeakyReLU | No |
| 2 | CONV-(N128,K3,S2), LeakyReLU | No |
| 3 | CONV-(N256,K3,S2), LeakyReLU | No |
| 4 | RESBLK-(N512,K1,S1) | No |
| 5 | RESBLK-(N512,K1,S1) | No |
| 6 | RESBLK-(N512,K1,S1) | No |
| $\mu$ | RESBLK-(N512,K1,S1) | Yes |

| Layer | Generators | Shared? |
|---|---|---|
| 1 | RESBLK-(N512,K1,S1) | Yes |
| 2 | RESBLK-(N512,K1,S1) | No |
| 3 | RESBLK-(N512,K1,S1) | No |
| 4 | RESBLK-(N512,K1,S1) | No |
| 5 | DCONV-(N256,K3,S2), LeakyReLU | No |
| 6 | DCONV-(N128,K3,S2), LeakyReLU | No |
| 7 | DCONV-(N64,K3,S2), LeakyReLU | No |
| 8 | DCONV-(N3,K1,S1), TanH | No |

| Layer | Discriminators | Shared? |
|---|---|---|
| 1 | CONV-(N64,K3,S2), LeakyReLU | No |
| 2 | CONV-(N128,K3,S2), LeakyReLU | No |
| 3 | CONV-(N256,K3,S2), LeakyReLU | No |
| 4 | CONV-(N512,K3,S2), LeakyReLU | No |
| 5 | CONV-(N1024,K3,S2), LeakyReLU | No |
| 6 | CONV-(N1,K2,S1), Sigmoid | No |

# B Domain Adaptation

**MNIST↔USPS.** For the MNIST and USPS adaptation experiments, we used the entire training sets in the two domains (60000 training images for MNIST and 7291 training images for USPS) for learning and reported the classification performance on the test sets (10000 test images for MNIST and 2007 test images for USPS). The MNIST and USPS images had different sizes, we resized them to 28×28 for facilitating the experiments. We trained the Coupled GAN algorithm [17] using the same setting for a fair comparison.

The encoder and generator architecture was given in Table 4. For the discriminators and classifiers, we used an architecture akin to the one used in the Coupled GAN paper, which is given in Table 5.

**SVHN→MNIST.** For the SVHN and MNIST domain adaptation experiment, we used the extra training set (consisting of 531131 images) in the SVHN dataset for the source domain and the training set (consisting of 60000 images) in the MNIST dataset for the target domain as in the DTN work [26]. The test set consists of 10000 images in the MNIST test dataset. The MNIST images were in gray-scale. We converted them to RGB images and performed a data augmentation where we also used the inversions of the original MNIST images for training. All the images were resized to 32×32 for facilitating the experiment. We also found spatial context information was useful. For each input image, we created a 5-channel variant where the first three channels were the original RGB images and the last two channels were the normalized x and y coordinates.

The encoder and generator architecture was the same as the one used in the MNIST↔USPS domain adaptation. For the discriminators and the classifier, we used a network architecture akin to the one

used in the DTN paper. The details are given in Table 6. We used dropout at every layer in the classifier for avoiding over-fitting.

Table 4: Encoder and generator architecture for MNIST↔USPS domain adaptation.

| Layer | Encoders | Shared? |
|---|---|---|
| 1 | CONV-(N64,K5,S2), BatchNorm, LeakyReLU | No |
| 2 | CONV-(N128,K5,S2), BatchNorm, LeakyReLU | Yes |
| 3 | CONV-(N256,K8,S1), BatchNorm, LeakyReLU | Yes |
| 4 | CONV-(N512,K1,S1), BatchNorm, LeakyReLU | Yes |
| $\mu$ | CONV-(N1024,K1,S1) | Yes |

| Layer | Generators | Shared? |
|---|---|---|
| 1 | DCONV-(N512,K4,S2), BatchNorm, LeakyReLU | Yes |
| 2 | DCONV-(N256,K4,S2), BatchNorm, LeakyReLU | Yes |
| 3 | DCONV-(N128,K4,S2), BatchNorm, LeakyReLU | Yes |
| 4 | DCONV-(N64,K4,S2), BatchNorm, LeakyReLU | No |
| 5 | DCONV-(N3,K1,S1), TanH | No |

Table 5: Discriminator architecture for MNIST↔USPS domain adaptation.

| Layer | Discriminators | Shared? |
|---|---|---|
| 1 | CONV-(N20,K5,S1), MaxPooling-(K2,S2) | No |
| 2 | CONV-(N50,K5,S1), MaxPooling-(K2,S2) | Yes |
| 3 | FC-(N500), ReLU, Dropout | Yes |
| 4a | FC-(N1), Sigmoid | Yes |
| 4b | FC-(N10), Softmax | Yes |

Table 6: Discriminator architecture for SVHN→MNIST domain adaptation.

| Layer | Discriminators | Shared? |
|---|---|---|
| 1 | CONV-(N64,K5,S1), MaxPooling-(K2,S2) | No |
| 2 | CONV-(N128,K5,S1), MaxPooling-(K2,S2) | Yes |
| 3 | CONV-(N256,K5,S1), MaxPooling-(K2,S2) | Yes |
| 4 | CONV-(N512,K5,S1), MaxPooling-(K2,S2) | Yes |
| 5a | FC-(N1), Sigmoid | Yes |
| 5b | FC-(N10), Softmax | Yes |