[Reviews · NeurIPS 2017]

Reviewer 1



The paper provides an elegant model framework to tackle the problem of image-to-image translation (unsupervised). The model framework consists of a VAE connected to a GAN, with weight sharing at the latter layers of the encoder and former layers of decoder. I've read and reviewed hybrid VAE-GAN approaches before, and this one seems a bit nicer than the rest. They also add skip-connections and stuff to cope with the spatial information loss in the code given to the decoder when you make the encoder deeper. In terms of experiments, they have three sets of experiments. They are sensible experiments, well constructed and all, but the results aren't shocking or much better than what I'd expect. Overall, a well-written, and elegant paper. Definitely a strong accept from my side.

Reviewer 2



This paper investigated the problem of unsupervised image-to-image translation (UNIT) using deep generative models. In contrast to existing supervised methods, the proposed approach is able to learn domain-level translation without paired images from two domains. Instead, the paper proposed to learn a joint distribution (two domains) from a single shared latent variable using generative adversarial networks. Additional encoder networks are used to infer the latent code from each domain. The paper proposed to jointly train the encoder, decoder, and discriminator in an end-to-end style without paired images as supervision. Qualitative results on several image datasets demonstrate good performance of the proposed UNIT pipeline. == Qualitative Assessment == The unsupervised domain transfer is a challenging task in deep representation learning and the proposed UNIT network demonstrated a promising direction based on generative models. In terms of novelty, the UNIT network can be treated as an extension of coGAN paper (reference 17 in the paper) with joint training objectives using additional encoder networks. Overall, the proposed method is sound with sufficient details including both qualitative and quantitative analysis. I like Table 1 that interpolates and compares the existing work as subnetworks in UNIT. Considering the fact that UNIT extends coGAN paper, I would like to see more discussions and explicitly comparisons between coGAN and UNIT in the paper. Performance-wise, there seems to be slight improvement compared to coGAN (Table 3 in the paper). However, the evaluation is done on MNIST dataset which may not be conclusive at all. Also, I wonder whether images in Figure 8 are cherry-picked? Compared to coGAN figures, the examples in Figure 8 have less visual artifacts. I encourage authors to elaborate this a little bit. * What’s the benefit of having an encoder network for image-to-image translation? Image-to-image translation can also be applied to coGAN paper: (1) analysis-by-synthesis optimization [1] & [2]; or (2) train a separate inference network [3] & [4]. Can you possibly comment on the visual quality by applying optimization-based method to a pre-trained coGAN? I encourage authors to elaborate this a bit and provide side-by-side comparisons in the final version of the paper. * What’s the benefit of joint training? As being said, one can train a separate encoder/inference network. It is not crystal clear to me why VAE type of training is beneficial. Maybe the reconstruction loss can help to stabilize the training a little bit but I don’t see much help when you add KL penalty to the loss function. Please elaborate this a little bit. Additional comments: * Consider style-transfer as alternative method in unsupervised image-to-image translation? Given the proposed weight sharing architecture, I would assume the translated image at least preserves the structure of input image (they are aligned to some extent). If so, style-transfer methods (either optimization based or feed-forward approximation) can serve as baseline in most experiments. * Just for curiosity, other than visual examination, what is the metric used when selecting models for presentation? Though the hyperparameters have little impact as described in ablation study, I wonder how did you find the good architecture for presentation? At least quantitative evaluations can be used as metric In MNIST experiment for model selection. Since you obtained good results in most experiments, I wonder if this can be elaborated a bit. [1] Generative Visual Manipulation on the Natural Image Manifold, Zhu et al. In ECCV 2016; [2] Reference 29 in the paper; [3] Picture: A Probabilistic Programming Language for Scene Perception, Kulkarni et al. In CVPR 2015; [4] Generative Adversarial Text-to-Image Synthesis, Reed et al. In ICML 2016;

Reviewer 3



The author(s) propose an architecture for un-paired correspondence learning between two domains by leveraging a combination of VAEs, GANs, and weight sharing constraints. The paper is clearly written with a nice suite of both quantitative and qualitative experiments. The improvement in Unsupervised Domain Adaptation on the SVHN->MNIST task from 84.88% to 90.53% is a strong result and much more interesting than the more trivial case of the MNIST->USPS and vice-versa since SVHN and MNIST are significantly different domains. The ablation experiments are an important inclusion as well since there are quite a few design/architecture decisions involved here. It would be a nice extension to see these ablation studies also carried out on the SVHN->MNIST task as the euclidean distance in pixel space metric which the current ablation test uses can be an unreliable and difficult to interpret as it has semi-arbitrary scaling. There have been quite a few related approaches in this area of research recently. While the authors include a paragraph in the related work section briefly discussing this, a more thorough comparison/discussion of the differences and motivations of their approach would be appreciated. In particular, the approach appears to have design decisions, such as high-level weight sharing, that are similar to "Coupled generative adversarial networks" which appeared at NIPS last year. A more clear comparison and discussion of the differences here could help readers better understand the contributions of the paper. Could the author(s) comment on what are the benefits/drawbacks of their approach compared to these several other architectures so as to help practitioners/researchers make better informed decisions in this space?